# Pacemaker Lead Endocarditis Investigated with Intracardiac Echocardiography: Factors Modulating the Size of Vegetations and Larger Vegetation Embolic Risk during Lead Extraction

**DOI:** 10.3390/antibiotics8040228

**Published:** 2019-11-19

**Authors:** Carlo Caiati, Paolo Pollice, Mario Erminio Lepera, Stefano Favale

**Affiliations:** Unit of Cardiovascular Diseases, Department of Emergency and Organ Transplantation, University of Bari, 70123 Bari, Italy

**Keywords:** infective endocarditis, lead vegetations, intracardiac echocardiography, embolism, pacemaker

## Abstract

Lead pacemaker infection is a complication on the rise. An infected oscillating mass attached to the leads (ILV) is a common finding in this setting. Percutaneous extraction of the leads and of the device is the best curative option. However, extraction of leads with large masses can be complicated by pulmonary embolism. The aim of this study was to understand the factors associated with large ILV using a sophisticated ultrasound technique to visualize the masses, namely intracardiac echocardiography (ICE), and investigate whether larger masses induce more complications during and after extraction. Percutaneous lead extraction and peri-procedural ICE were done in 36 patients (pts) (75 ± 11 years old, 74% males). Vegetations (max dimension = 8.2 ± 4.1 mm) in the right cavity were found in 26 of them, mostly adhering to the leads. We subdivided the patients into 2 groups: with vegetation size < 1 cm (18 pts) and vegetation size ≥ 1 cm (8 pts). By univariate analysis, we found that patients in group 1 were more often taking anticoagulation therapy (*p* = 0.03, Phi (Phi coefficient) = −0.5, OR [odds ratio] 0.071) and had signs of local pocket infection (*p =* 0.02, Phi = −0.52, OR 0.059) while significantly more patients in group 2 had diabetes (*p* = 0.08, Phi = 0.566, OR 15); moreover the patients in group 2 showed a trend toward a more frequent positive blood culture (*p* = 0.08, Phi = 0.39, OR 5.8) and infection with coagulase negative staphylococci (*p =* 0.06, Phi = 0.46, OR 8.3). At multivariate analysis, only 3 factors (diabetes, younger age and anticoagulation therapy) were independently associated with ILV size: diabetes, associated with larger vegetations (group 2), showed the largest beta value (0.44, *p* = 0.008); age was inversely correlated with ILV size (beta value = −32, *p =* 0.038), and anticoagulation therapy (beta value = −029, *p* = 0.048) was more commonly associated with smaller vegetations (group 1). Larger ILV were not associated with more complications or death during or after the extraction. Conclusion: diabetes, anticoagulation therapy and age are independent predictors of lead vegetation size. The embolic potential of large ILV during extraction was modest, so ILVs >1cm are not a contraindication to percutaneous extraction of infected leads.

## 1. Introduction

Pacemaker lead infection is a very serious medical complication, and is unfortunately on the rise [1,2]. This condition is associated with substantial mortality, morbidity and financial cost [3]. The criteria for detecting this complication are the combination of clinical signs like fever, malaise, lung signs of infection etc., along with (1) isolation of microbes in the blood, (2) detection of an oscillating mass attached to the lead by imaging techniques like transesophageal and intracardiac echocardiography, and finally, (3) the presence of a device pocket infection [4]. 

In the presence of such infections, the lead(s) extraction procedure is the more radical intervention, generally attained by percutaneous approach using different techniques [5,6,7]. However, the inherent risk of embolization during the percutaneous extraction procedure is enhanced by the size of the mass affecting the leads [8,9,10]. It is common practice to treat patients with antibiotics for an additional period of time in the hope of shrinking the mass and thus performing the percutaneous extraction procedure under safer conditions. Big masses (>1 cm), and certainly those larger than 2–3 cm, in fact, can embolize to the lungs during the extraction, creating significant pulmonary arterial bed obstructions and/or pulmonary abscess [10]. 

However, the factors causing enlargement of the infected mass are not clearly understood, as few retrospective studies are available, and the findings are not concordant [11,12,13]. Therefore, interventions aimed at reducing the mass size are done empirically and with unpredictable results: most often, the mass does not change after such empirical treatments. The few studies available indicate that larger infected masses could be associated with non-modifiable factors like renal failure, heart failure, implantable cardioverter defibrillator, and loops of the lead or a lead abrasion assessable only after extraction [13]. In addition, in these studies, imaging of the infected mass was performed with a sub-optimal ultrasound approach like transthoracic echo, which is not conclusive for lead mass evaluation in the right heart, or transesophageal echocardiography, which also has limitations with respect to imaging of the right heart since penetration of ultrasounds from the esophagus can be hampered by lipomatous infiltration of the atrial septum [14] and any kind of atrial septal thickening, which are not infrequent findings in the elderly. 

Intracardiac echocardiography (ICE) has a superior capacity to visualize lead abnormalities such as lead fibrotic thickening and lead fibrotic adherence to the cardiac wall, never before investigated in vivo, as was recently demonstrated [15]. It is also better than transesophageal echocardiography, showing the presence of even small oscillating vegetations attached to the lead in cases of cardiac device-related endocarditis (CDIE) [16]. Lead imaging by ICE virtually excludes the risk of missing masses or making wrong measurements of masses; this more precise evaluation of the mass is important for a more reliable evaluation of the association of mass size to possible independent predictors. So we hypothesized that the precise evaluation of lead vegetations by ICE performed immediately before the extraction procedure could help to understand the factors modulating the size of the infected mass attached to the pacemaker leads and the embolic potential of larger masses during lead extraction. For this purpose, 36 patients with infected pacemaker leads were prospectively studied with ICE before the percutaneous leads extraction procedure. 

## 2. Results

Among the 36 patients with CDIE, we found only 1 major CDIE criterion as defined by Sohail [4] in 7 (19%) patients, namely a device pocket infection, 2 criteria in 21 patients (58%), and 3 in 8 (22%). Among minor criteria, we had only 1 patient with an embolic episode and 7 patients with fever. So a definite diagnosis of CDIE was possible in all but these last 7 cases, which had only pocket infection and an insufficient number of other minor criteria. Thus, classic general signs and peripheral embolism, unlike left side endocarditis [17], were almost completely lacking. 

Twenty-six of 36 patients (72%) had vegetation masses detected by ICE (maximal and minimal mass sizes were 8.5 ± 4 mm and 3.3 ± 2.43 mm, respectively) (Figure 1). This group was subdivided on the basis of vegetation size into subgroup 1 (18 patients with small vegetations: size < 1 cm) and subgroup 2 (8 patients with large vegetations: size ≥ 1 cm) (Table 1). Patient demographics are summarized in Table 1, and examples of small and large vegetations are presented in Figure 2 and Figure 3, respectively.

### 2.1. Mass Features and Associations

The vegetations visualized by ICE appeared as very mobile masses and were attached mostly to the leads; however, in 3 cases masses were not attached to the working leads but to other intracardiac structures or to abandoned leads; such unusual locations (specifically the right atrial wall, right atrial appendage and the abandoned ventricular lead at its crossing with the other ventricular lead) were significantly more common in group 2 (*p* < 0.05). 

Univariate analysis (Table 1). In group 1, the patients were older, but this was not significant at t-test. However, reanalyzing by one-way ANOVA after collapsing age in 3 groups (≤ 68 years, 69–82 and 83+), the group of younger patients had significantly larger vegetations (12 ± 4 mm) than the 69–82-year-old group (5.4 ± 1.2 mm, *p* < 0.001). Diabetics were significantly more frequent in group 2 than non-diabetics (67% vs. 12% odds ratio [OR] =15 [95% CI = 2.0–113], *p* < 0.008 and the effect size of this association was strong, phi = 0.57); patients that had been taking anticoagulation therapy before the ICE examination (13 patients: 8 [31%] on warfarin, 4 [15%] on rivaroxaban and 1 [4%] on low molecular weight heparin) showed a significant association with a smaller mass (92% vs. 46%, *p* = 0.03). A positive blood culture suggesting a remote infection source seeding onto the leads via the blood, either originating from the device pocket or from another remote site, was associated with the larger vegetations size, albeit only with marginal significance (*p* = 0.08) (Table 1). Not unexpectedly, this factor was also more commonly associated with clinical signs of infection like fever (38% vs 9%, *p* < 0.07). The presence of a device pocket infection was significantly associated with a smaller size vegetation (81% vs. 19%, phi = −0.52, *p* < 0.02) and a positive device pocket swab was even more strongly associated with the smaller vegetation size (89% vs. 25%, *p* = 0.003, phi = 0.63).

Blood culture bacteriology. The most frequently isolated strain was the staphylococcus coagulase negative bacterium (such as Staphylococcus hominis, epidermidis and streptococcus anginosus) (in 8 out of 26 patients, 31%), while in only one case (3.8%) was staphylococcus aureus found; in only one case, a rare infection with candida albicans was documented [18]. At univariate analysis staphylococcus coagulase negative infection tended to be more frequent in the group with large vegetations (group 2) (*p* = 0.06) (Figure 3) (Table 1).

Multiple regression was applied to assess the ability of the most significant variables with the higher effects size to predict mass size lead endocarditis at univariate analysis. Since the sample size was limited, we used only few variables. The total variance explained by the model as a whole was large, being 57% (adjusted R square); in the final model, only diabetes, anticoagulation therapy pre-extraction, and age were statistically significant predictors; diabetes showed a higher beta value (0.44 *p* < 0.008) than age (−32.3, *p* < 0.038) and anticoagulation pre-extraction (−0.29, *p* < 0.048). 

Interactions effects. We found significant interaction effects first between anticoagulation and diabetes in predicting mass size: F-test (1, 22) = 10.168, *p* = 0.004, with large effect size (partial eta squared = 0.31). The significant interaction between diabetes and anticoagulation indicates that the anticoagulation effect on the mass size depends in part upon the diabetic status: only when diabetes was present was the anticoagulation effective in reducing the mass size in the treated group; on the other hand, in patients without diabetes, the anticoagulation had no effect in reducing the mass size (Figure 4). The second significant interaction was between coagulase negative staphylococcus and fever in predicting mass size: F (1.22) = 9.189, *p* = 0.006, with large effect size (partial eta squared = 0.29). There was no statistically significant main effect for fever and coagulase negative staphylococcus variables, in accordance with the previously reported univariate analysis (Table 1). This interaction means that the association of coagulase negative staphylococcus with a large mass depends on fever: if blood culture is positive for staphylococcus coagulase it predicts a large mass only in the absence of fever (that occurred in most cases) (Figure 5).

### 2.2. Follow-up

Six patients had clinical adverse events during follow-up after the lead extraction procedure, all within one month of the extraction procedure: 3 died (one of acute respiratory distress syndrome possibly due to pulmonary embolism, one of multiple organ failure secondary to multiple antibiotic toxicity, and the last of acute heart failure) and 3 had short living dyspnea associated with pulmonary infiltrates at the chest x-ray. These 6 adverse events were not associated with vegetation mass size: four events occurred in group 1 (4 out of 18, 22.2%) and 2 in group 2 (2 out of 8, 25%), (chi square 0.024, *p* = ns). 

## 3. Discussion

The main finding in this report is that 3 variables are independently associated with vegetation size: firstly, diabetes type 2 (with a stronger predictive power) is associated with a larger mass size; secondly, anticoagulation therapy before extraction is associated with a small mass size (and vice versa, no anticoagulation therapy is associated with a larger mass size) and finally, patient age is negatively correlated with the mass size: the younger the patient the larger the vegetation in the course of CDIE.

In addition, in our study group, vegetation size was not associated with death or other grim clinical events after percutaneous extraction of the leads (*p* = ns [not statistically significant]). 

One advantage of this study over the previous ones [11,12,13] is that the mass size was very accurately assessed, using ICE, which has proven to be the gold standard for imaging pacemaker leads [15] and vegetation adhering to the leads [16]. In fact, ICE performed with a catheter equipped with a linear phased array multifrequency (5.5 to 10 MHz) steerable transducer with 4 steering directions is able to demonstrate even subtle strands as masses measuring a few mm and to visualize the lead along its entire course from the superior vena cava throughout the right ventricle. In addition, there is no other tissue that can cause ultrasound attenuation as occurs with transesophageal echocardiography since the TEE ultrasound plane must traverse the interatrial septum in order to insonify the right cavity. When the atrial septum is thickened by infiltration, like in lipomatous hypertrophy of the atrial septum [14] (quite frequent in the elderly) or amyloid infiltration [19] the ultrasound transmission is attenuated and so inadequate for appropriate insonification and thus visualization of the right cavity and of the pacemaker leads. In severe BPCO, too, the esophageal window may be restricted owing to hyperinflation of the lungs and the interposition of air impeding the ultrasound beam directed toward the right cardiac chambers. 

Diabetes is a prime risk factor for cardiovascular disease [20]. In particular, it is a major known con-causal etiologic factor in causing coronary artery disease [21]. In our study, diabetes was also associated with larger vegetations. Numerous studies have shown that coagulation abnormalities occur in the course of diabetes mellitus, resulting in a state of thrombophilia. The abnormalities observed involve all stages of coagulation, affecting both thrombus formation and its inhibition, fibrinolysis, platelet and endothelial function. For this reason, diabetes can be defined as a hypercoagulable state [22]. This is very relevant to vegetation formation. In fact, the factors predisposing to vegetations in the right heart are leads, central venous catheters, drug abuse and heart defects [23], all factors that activate the coagulation cascade as they are either foreign bodies (like leads, catheters) or because they induce micro damage of the endocardium through intra-cardiac jet abnormalities owing to congenital, valvular or dynamic intracardiac obstruction abnormalities. Regarding leads, one landmark animal study [24] demonstrated that leads induce endothelial damage (of the right endocardial surface) and flow perturbation around the lead; this brings about thrombosis [24] both on the surface of the leads and on the damaged endocardium. Then thrombosis can be dissolved or become organized. So the leads create a dynamic state of microthrombosis on their surface and on the lead-injured endocardium. Such lead-induced thrombosis with platelet deposition and fibrin is the first of 4 events bringing about infective lead endocarditis [25,26] by creating a non-bacterial thrombotic vegetation (NBTV). So if there is bacteremia (2nd event), adherence of micro-organisms to the NBTV takes place, with subsequent deposition of more fibrin and platelets that cover the infective agents (3rd event) favoring their multiplication (4th event) [25,26]. Thus, diabetes is a kind of enhancer of both nonbacterial and bacterial thrombotic apposition to the leads. In addition, sterilization of the ILV takes place thanks to the effective action of leukocytes and in particular polymorphonuclear leukocytes. This immunologic reaction is impaired in diabetics, so the microbes live longer, thereby creating more platelet adherence and thrombosis [27]. 

Anticoagulation therapy was significantly associated with small masses (Figure 2). This is not unexpected. In fact, after inoculation of bacteria in animals with catheters in the heart, grossly visible vegetation was prevented by the administration of anticoagulants [28], and anticoagulant treatment protected against the development of infection in a large retrospective study of 1837 patients [29]. On the basis of our findings, anticoagulation should be promptly instituted in order to reduce the vegetation mass size but only after an effective antibiotic treatment has been initiated. In fact, if anticoagulation is administered but the bacterial burden is not eliminated, we may even worsen the course of CDIE, as demonstrated in animals [28,30] despite a reduced mass size. Moreover, we have demonstrated with a statistical interaction test that anticoagulation can be much more effective to reduce the mass size in diabetics (Figure 4). This can find an explanation in the hypercoagulable status of diabetics [22], which enhances the formation of both bacterial and nonbacterial thrombi, so that the anticoagulation effect at preventing thrombus growth is much more evident than in the non-anticoagulated diabetic group. The prevalent thrombotic nature of this infected masses affecting the leads, which confirms the important role of anticoagulant therapy, as also sustained by the potential utility of thrombolysis to reduce very large masses (> 4 cm) [31]. 

Our findings do not show chronic renal failure associated with large vegetation masses, in contrast to previous papers [11,13,32]. This can be explained by the mild grade of chronic renal failure in our patients (only 4 patients [15%] had a glomerular filtration rate < 60), and none on dialysis, unlike the other reports that included patients with more severe kidney dysfunction (dialysis) in the analysis [11].

In our study, crossing of multiple leads, especially in a slow flow chamber like the right atrium, can create larger masses, and this is attributable to lead abrasion owing to friction with the catheter, confirming the potential role of lead abrasion in creating larger vegetations, as previously indicated [13].

Vegetations and isolated bacterial strains. We found that larger vegetations (Figure 3) tended to be significantly associated with coagulase negative staphylococci. More indolent organisms such as coagulase-negative staphylococci, shielded to the immune system by the film that they produce [33], can be associated with less systemic signs of infection and this brings about a slower disease progression and hence later assessment of larger vegetations. This interpretation is confirmed by our own findings regarding a significant coagulase negative staphylococci interaction with fever: the absence of fever (torpid reaction of the immune system to these film-protected germs) makes this germ strain a significant predictor of a large mass (Figure 5). Our findings are perfectly in agreement with a recent report of the MEDIC study (Multicenter Electrophysiologic Device Cohort) [12]. 

### 3.1. Clinical Implications

Accurate vegetation detection and size assessment aids clinical decision-making in CDIE patients. Large masses (> 1 cm) can cause embolization and increase morbidity and potential mortality after lead extraction. In our series, however, the event rate was not associated with vegetation size. In any case, our “large” infected masses were no more of 18 mm in max dimension (Figure 1), so relatively small, limiting the embolic potential during percutaneous lead extraction. Very large masses >3 cm can pose a serious risk for pulmonary embolization [10,31]. However, on the basis of our data, we also believe that prompt institution of appropriate anticoagulation is an effective approach to achieve mass shrinkage, but only after appropriate antibiotic therapy has been started. In addition, we have shown that some modifiable factors are associated with vegetation size (Table 1). In particular, diabetes can make the infection worse so prompt treatment of diabetes, especially with an appropriate diet that is the sole etiologic intervention, may ameliorate this lead infection and hopefully cause a regression or a shrinkage of the mass.

Finally, this study is in agreement with and corroborates the few other studies [12,13] reporting the association of pocket device infection with small mass size, the negative correlation of age with mass size, and the positive relationship of the mass size with coagulase negative staphylococci, and in general with positive blood culture even though the later 2 variables were only marginally statistically significant in our study. 

### 3.2. Limitations

The major study limitation is the small sample size (only 26 cases with masses), which reduces the statistical power. This problem, however, is outbalanced by the more precise evaluation of the mass size by a more sophisticated imaging technique (ICE) which, by reducing the random error in measurement, can reduce the need for a very large sample size in assessing association. This is supported by the fact that our data are in agreement with other larger studies [11,12,13] in which less sophisticated imaging technique were adopted for intracardiac mass visualization.

## 4. Materials and Methods 

In our exploratory, single center prospective study, we enrolled 36 patients from January 2014 to September 2017, referred to our department for lead(s) extraction due to infective endocarditis. All data concerning clinical and microbiological status of the patients were recorded before the extraction procedure. 

All enrolled patients underwent intracardiac echocardiography during the procedure. ICE was performed twice during the extraction procedure: before extraction to gain a pre-extraction picture of the catheter and immediately after extraction. All intracardiac examinations were performed by one experienced cardiologist (CC). 

### 4.1. Clinical Management 

During hospitalization, all patients began antibiotic therapy; wound swab culture and blood cultures were performed before (preferably in wash-out from antibiotic therapy) and after the extraction; lead(s) tip culture was performed after the procedure. 

In the pre-extraction phase, cardiac device-related infective endocarditis (CDIE) was defined in accordance with Sohai [4], We considered as major criteria for LAE (1) ICE detection of the mass, (2) positive blood culture, (3) device pocket infection with positive swab culture, (4) positive lead culture in the absence of microbiological or clinical evidence of device pocket infection (since all patients underwent percutaneous extraction of the device that can contaminate the tip of the lead during its course through the pocket). As minor criteria we considered (1) fever > 38°, vascular findings (arterial pulmonary embolus, septic pulmonary infarct, pulmonary infiltrates at chest x-ray), immunological findings (arthralgia, back pain with signs of spondylitis, glomerulonephritis, Osler nodules, Roth’s spot) [34]. Diagnosis of definite CDIE was made in the presence of at least 2 major or 1 major and 3 minor criteria. 

Accordingly, the above validated criteria and major bacteriologic criteria were defined by positive blood cultures for typical endocardial pathogens or persistently positive for a micro-organism consistent with infective endocarditis [34,35]. Local device infection was clinically defined by the presence of local signs of erythema, warmth, fluctuance, erosions, tenderness, and purulent discharge [36].

### 4.2. Intracardiac Echocardiography 

We used catheters (8 Fr, Acunav^TM^ Siemens, Munich, Germany) equipped with a linear phased array multifrequency (5.5 to 10 MHz) steerable transducer with 4 steering directions: antero/posterior and left/right deflection of the catheter. The transducer produces a longitudinal scanning plane with a 90° sector image and has a tissue penetration capability of 15 mm; it was connected to a Siemens, either Acuson P50 portable ultrasound system (Siemens, Munich, Germany) or Acuson Sequoia 512 Ultrasound System (Siemens, Munich, Germany). 

Before extraction, the Acunav^TM^ catheter (length of 90 cm) was inserted into a femoral vein and, under fluoroscopy guidance, advanced through the inferior cava vein up to the superior vena cava; then images were obtained, first at the level of the superior vena cava and then while appropriately withdrawing, at the level of the right atrium. At this level it was rotated and tilted up and down in order to visualize the catheter(s) and all atrial segments, including the right appendage; then in rare cases it was curved and advanced into the right ventricle to better explore beneath the tricuspid valve; in all these regions, ICE allowed accurate visualization of the leads and of the right cardiac structures, as previously described [16]. 

To allow the extraction procedure the Acunav was withdrawn down to the inferior vena cava. After the extraction the catheter was repositioned in the superior vena cava and the imaging sequence as described above was obtained at this second time.

### 4.3. End Points

We specifically aimed to visualize mass-related endocarditis in the pre-extraction phase. 

Endocarditis vegetation was defined as an oscillating intracardiac mass on the electrode lead(s), cardiac valve leaflets or endocardial surface, confirmed by imaging on more than 1 echographic plane, in cases of valve or lead infection identified by positive blood or lead-tip cultures [34,37,38]. We collected information about the localization, shape and size of the vegetations and in case of multiple masses the biggest one was used for analysis.

All patients underwent clinical follow-up (mean follow-up time = 22 ± 12 months), focused on death, pulmonary embolism, infectious events, fever.

### 4.4. Statistical Analysis

Continuous variables were expressed as arithmetic means and standard deviation. Dichotomous variables were expressed as number and percentage. The Shapiro–Wilk test was used to test for normal data distribution. 

The study population was divided into two groups: (i) those with small vegetations (max diam <1 cm) and (ii) those with large vegetations (max diam ≥ 1 cm), using the largely accepted cutoff of 1 cm [12]. The utility of this cutoff was also basically confirmed by a clustering analysis (nonhierarchical cluster analysis for mixed data Two Steps algorithm) [39]. Differences between Groups 1 and 2 were analyzed using Student’s t-test for continuous variables and either Yates’ χ^2^ test or Fisher’s exact test as appropriate for dichotomous variables; a Phi coefficient (0.10 indicates small effects, 0.3 medium effect and 0.5 large effect) was calculated to assess the effect size statistics in 2 by 2 tables. Univariate and linear multivariate regression analyses were performed to estimate the relationship between risk factors and vegetation occurrence. Parameters reaching a significance level of *p* < 0.05 in the univariate analysis were entered into a multivariate regression analysis.

Interaction effects between the main variables reported in Table 1 were assessed by two-ways between-groups ANOVA and Eta squared test was calculated for the effect size with ANOVA (0.01 indicates small effect, 0.06 medium effect and 0.138 large effect). 

Statistical calculations were performed using IBM SPSS statistics version 23, Armonk, NY, USA: IBM Corp.

### 4.5. Ethics Approval and Consent to Participate

The study protocol was approved and provided by the Policlinico di Bari, Bari, IT Institutional Review Board (104/C.E.). All participants provided written informed consent. 

## 5. Conclusions

Diabetes, anticoagulation therapy and age are independent predictors of lead vegetation size. The embolic potential of large masses during extraction is modest so does not pose a contraindication to percutaneous extraction of the infected leads. 

## Figures and Tables

**Figure 1 antibiotics-08-00228-f001:**
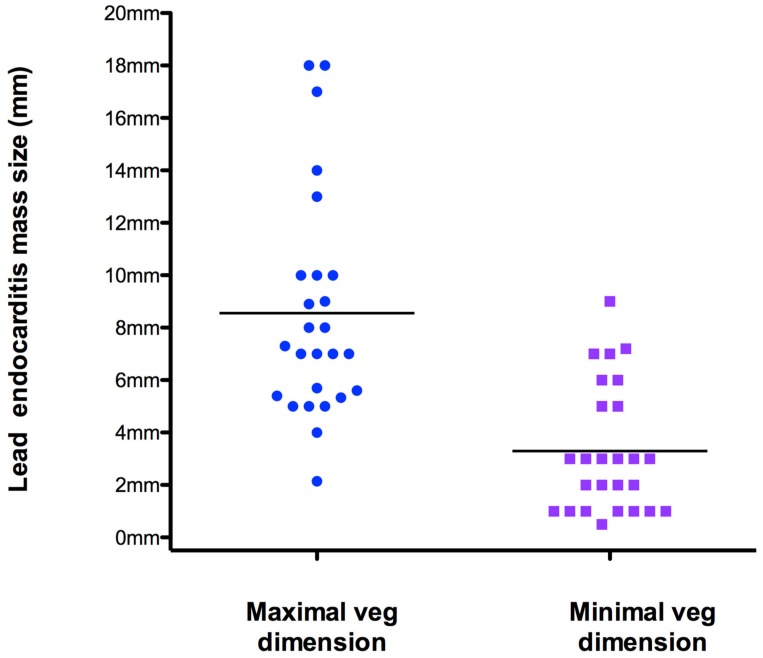
Individual value bar-graph depicting longer and shorter dimensions of lead vegetations. The two lines in the graph indicate the mean values. Veg = vegetation.

**Figure 2 antibiotics-08-00228-f002:**
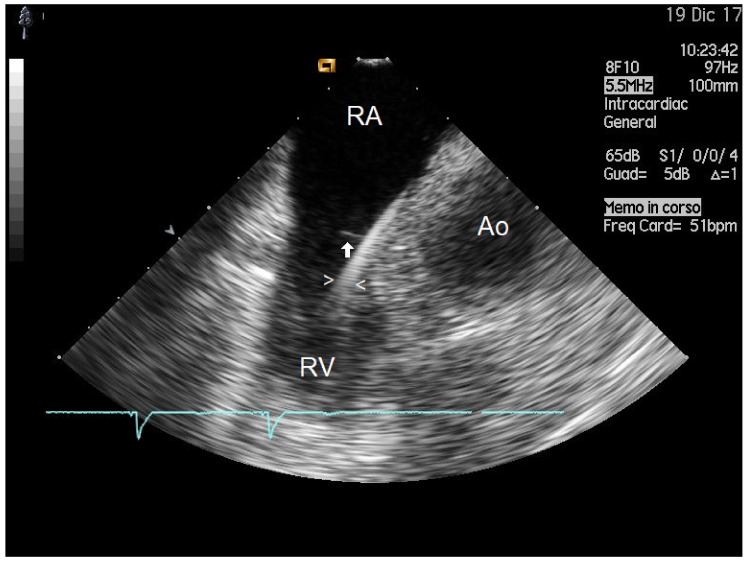
Lead infective endocarditis of smaller sizes. Example of a very small-sized mass attached to the lower-atrial portion of the lead (indicated by white arrowheads) of a single chamber PM, underscoring the high potential of ICE in visualizing even very small vegetations. This ICE “home view” projection transects the right atrium, tricuspid valve, right ventricle and ascending aorta. The vegetation (indicated by the white arrow) has a “strand” type morphology, an isoechogenic appearance, size 4 x 1 mm and high grade mobility. The patient underwent successful extraction of the infected device. (PM = pacemaker; RA = right atrium; RV = right ventricle; AO = aorta). At the bottom (in blue) electrocardiogram tracing of the patient, simultaneously recorded.

**Figure 3 antibiotics-08-00228-f003:**
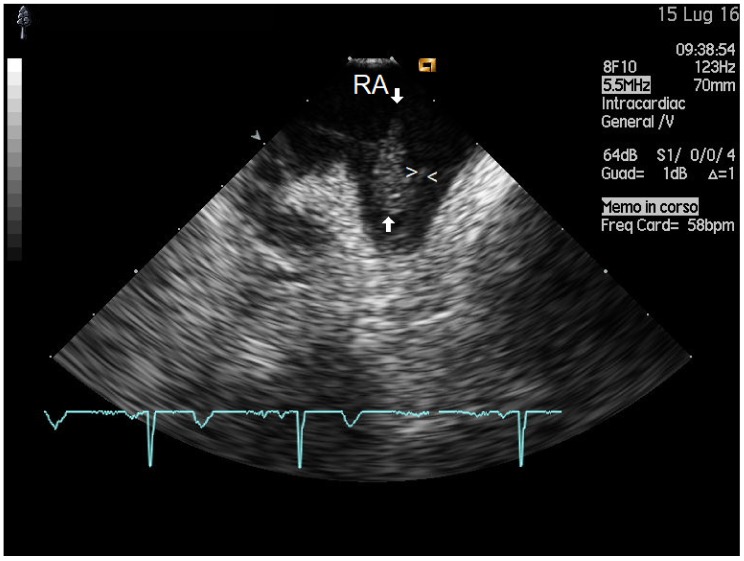
Lead infective endocarditis of larger sizes. Example of a mass attached to the terminal portion of an atrial catheter (indicated by white arrowheads) before its implantation in the right atrial appendage wall, as visualized by a specific plane orientation that transects the right atrium and right atrial appendage by means of a slight retroflexion of the probe head from a home view. This demonstrates the high potential of ICE in visualizing cardiac structures that are not accessible with other imaging modalities. The vegetation (indicated by the white arrows) has a “round shaped pedicular growth” type morphology, an iso-hyperechogenic appearance, size 14 x 8 mm, and it is attached with a sessile base to the lead with a high grade of mobility. The patient underwent successful extraction of the infected device. ICE = intracardiac echocardiography; PM: implantable cardioverter defibrillator; RA: right atrium; at the bottom (in blue) electrocardiogram tracing of the patient, simultaneously recorded.

**Figure 4 antibiotics-08-00228-f004:**
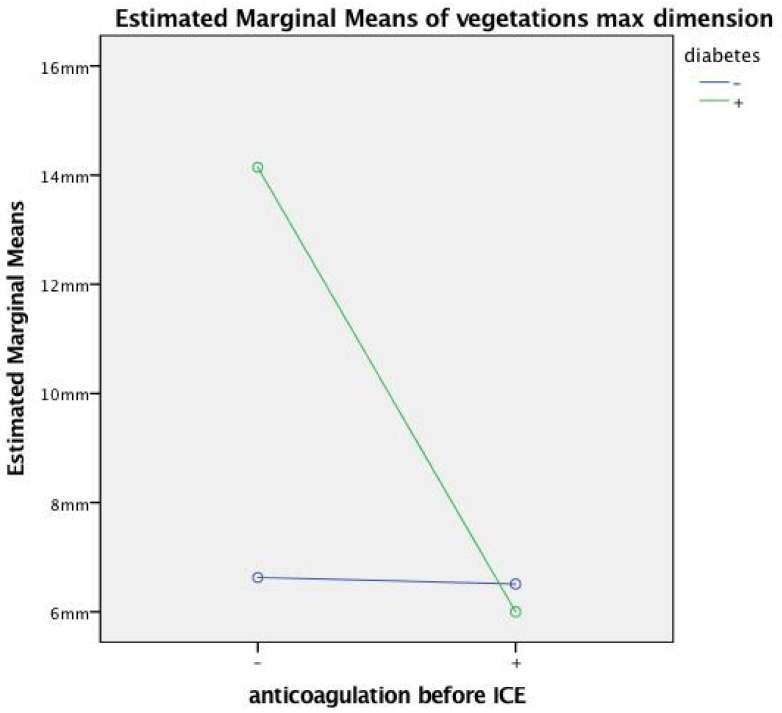
Plot illustrating diabetes and anticoagulation before ICE interaction by a two-way between-groups ANOVA. - = finding absent; + = finding present; ICE = intracardiac echocardiography.

**Figure 5 antibiotics-08-00228-f005:**
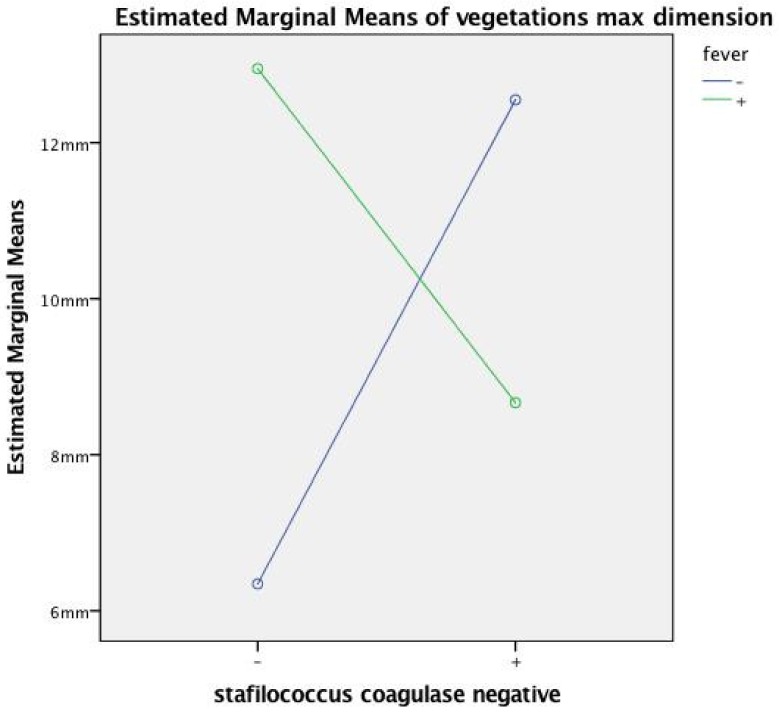
Plot illustrating fever and staphylococcus coagulase negative interaction in predicting vegetation size by a two-way between-groups ANOVA. - = finding absent; + = finding present; ICE = intracardiac echocardiography.

**Table 1 antibiotics-08-00228-t001:** Risk factors modulating lead vegetations size (univariate analysis).

Independent Variables		Group 1 (Veg < 1 cm)	Group 2 (Veg ≥ 1 cm)	OR (95%CI)	*p* Value
**Age**		75 ± 12	67 ± 14		ns
**Sex, #pts(%)**	male	17 (73.9)	6 (26.1)	1	ns
	female	1 (33.3)	2 (66.7)	5.67 (0.43–74.38)	ns
**BMI**		27.33 ± 4.9	29.21 ± 7.1	−	ns
**Cardiac Disease, #pts (%)**	prim DCM	7 (77.8)	2 (22.2)	−	ns
	isch DCM	6 (85.7)	1 (14.3)		
	Infilt HCM	0 (0.0)	1 (100)		
	Valv dis	1 (100)	0 (0.0)		
	prim arrhyt	4 (50)	4 (50)		
**Diabetes, #pts (%)**	−	15 (88.2)	2 (11.8)	1	= 0.008
	+	3 (33.3)	6 (66.7)	15 (2.0–113)	
**HF, #pts (%)**	−	10 (71.4)	4 (28.6)	1	ns
	+	8 (66.7)	4 (33.3)	1.2 (0.24–6.6)	
**EF**		45 ± 12	47 ± 10	−	ns
**CRF, #pts (%)**	−	13 (72.2)	5 (27.8)	1	ns
	+	5 (62.5)	3 (37.5)	1.6 (0.27–9.1)	
**Glomeral filtration rate(mL/m)**		79.33 ± 14	73.75 ± 34	−	ns
**Fever, #pts (%)**	−	15 (71)	6 (29)	1	ns
	+	3 (60)	2 (40)	1.7 (0.22–12.61)	
**Blood colture positive, #pts (%)**	−	14 (82)	3 (18)	1	= 0.08
	+	4 (44)	5 (56)	5.8 (0.95–35.72)	
**pocket infection, #pts (%)**	−	1 (20)	4 (80)	1	= 0.02
	+	17 (81)	4 (19)	0.059 (0.005–0.68)	
**Inf pocket colture positive, #pts (%)**	−	2 (25)	6 (75)	1	= 0.003
	+	16 (88.9)	2 (11.1)	0.042 (0.005–0.37)	
**CRP**		17 ± 36	12.41 ± 11	−	ns
**ESR**		29.4 ± 22	31.1 ± 25	−	ns
**#leads**		2.22 ± 0.6	2.13 ± 0.3	−	ns
**lead implantation time(m)**		128 ± 75	130 ± 74	−	ns
**COneg St, #pts** **(%)**	−	15 (83.3)	3 (16.7)	1	= 0.06
	+	3 (37.5)	5 (62.5)	8.3 (1.25–55.35)	
**Anticoag, #pts (%)**	−	6 (46.2)	7 (53.8)	1	= 0.03
	+	12 (92.3)	1 (7.7)	0.071 (0.007–0.72)	

# = number; pts = patients; OR = odds ratio; CI = confidence interval; BMI = body mass index; prim = primary; DCM = dilated cardiomyopathy; infilt = infiltrative; HCM = hypertrophic cardiomyopathy; Val dis = valvular disease; prim arrhyt = primary arrhythmias; HF = heart failure; CRF = chronic renal failure; Inf = infected; EF = ejection fraction; m = months; CRP = C reactive protein; ESR = erythrocytes sedimentation rate; COneg St = coagulase negative staphylococci; Anticoag = anticoagulation at the imaging evaluation; − = factor absent; + = factor present; Veg = vegetation; ns = not statistically significant

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
