# Peer review of "Pacemaker Lead Endocarditis Investigated with Intracardiac Echocardiography: Factors Modulating the Size of Vegetations and Larger Vegetation Embolic Risk during Lead Extraction"

_antibiotics, 2019, doi:10.3390/antibiotics8040228_

Round 1

Reviewer 1 Report

This is a nice study investigating factors responsible for larger vegetations in patients with pacemakers. However, there are few points that have to be addressed: 

Line 88 - there are errors referring to non-existent source. Correct it.  Although in Methods section it states significance level was considered at p<0.05, in the text differences with p value of 0.06 or 0.08 are described as significant.  Authors describe that patients in group 1 were more adherent to anticoagulation therapy but fail to provide any further details. This aspect (anticoagulation) is probably the most important in these patients determining the size of vegeations. Please describe what anticoagulants and/or antiplatelet regimen were these patients on.  Authors do not discuss whether there might be an interplay of the 'factors responsible for larger vegetations' identified in this study.  There is probably a glitch in the reference manager program used by the authors, few references are just missing. 

Reviewer 2 Report

  The paper is interesting and well-written.   However, please pay attention to some typos and some references sources which appear as not found.   Given the relevance of the topic, it would be suggested to broaden the analysis of literature.   With regards to the method, it would be of interest to consider the use of cluster analysis. In this concerns, there are some of the suggested researches that may be taken into consideration:   - Caruso G., Gattone S. A., Balzanella A. and Di Battista T. Cluster analysis: an application to a real mixed-type data set. In Models and Theories in Social Systems; Flaut, C., Hošková-Mayerová, Š., Flaut, D., Eds.; Springer International Publishing, Studies in Systems, Decision and Control: 2019; 179, pp. 525--533.   - Caruso G. and Gattone S. A. Waste management analysis in developing countries through unsupervised classification of mixed data. Social sciences, 2019.   Bucciarelli, E., Chen, S., Corchado, J. M., Eds.; Springer International Publishing, Advances in Intelligent Systems and Computing: 2018; 618, pp. 48--55.   I encourage the authors to refine their paper to make it certainly available for publication in the journal.    

Reviewer 3 Report

Cardiac device-related endocarditis (CDE) such as pacemaker lead has some clinical challenges. One is the patient who needs the implanted device, and the potential morbidity and mortality associated with its removal. Second one is, the problem of a persistent infection – usually developed during insertion of an electrical device – that is resistant to many antibiotics, and the complicated surgery to remove the device. Lead infection generally occurs due to hematogenous seeding of the organism in the absence of pocket infection. Risk factors such as diabetes mellitus and immunosuppression are usually present.

It is an important study. In the manuscript authors have demonstrated the association of various factors such as age, diabetes and anticoagulation therapy with the lead vegetation size and extraction.

There are limited studies on Cardiac device-related endocarditis with small number of patients.

Author Response

no response because the manuscript was OK for him/her

Round 2

Reviewer 1 Report

Authors addressed all the comments this reviewer had! 

One last thing - please seek professional English editing. There are numerous missuses of adverbs, articles and prepositions.